# Performance Influencing Factors of Convolutional Neural Network Models for Classifying Certain Softwood Species

**Jong-Ho Kim [1]**, **Byantara Darsan Purusatama [2]**, **Alvin Muhammad Savero [1]**, **Denni Prasetia [1]**, **Go-Un Yang [1]**, **Song-Yi Han [2]**, **Seung-Hwan Lee [1,2]** and **Nam-Hun Kim [1,\*]**

[1] Department of Forest Biomaterials Engineering, College of Forest and Environmental Sciences, Kangwon National University, Chuncheon 24341, Republic of Korea; kimjongho@kangwon.ac.kr (J.-H.K.); alvinsavero@kangwon.ac.kr (A.M.S.); dnnprsty98@kangwon.ac.kr (D.P.); rhdns7274@kangwon.ac.kr (G.-U.Y.); lshyhk@kangwon.ac.kr (S.-H.L.)

[2] Institute of Forest Science, Kangwon National University, Chuncheon 24341, Republic of Korea; byantara@kangwon.ac.kr (B.D.P.); songyi618@kangwon.ac.kr (S.-Y.H.)

\* Correspondence: kimnh@kangwon.ac.kr

**Abstract:** This study aims to verify the wood classification performance of convolutional neural networks (CNNs), such as VGG16, ResNet50, GoogLeNet, and basic CNN architectures, and to investigate the factors affecting classification performance. A dataset from 10 softwood species consisted of 200 cross-sectional micrographs each from the total part, earlywood, and latewood of each species. We used 80% and 20% of each dataset for training and testing, respectively. To improve the performance of the architectures, the dataset was augmented, and the differences in classification performance before and after augmentation were compared. The four architectures showed a high classification accuracy of over 90% between species, and the accuracy increased with increasing epochs. However, the starting points of the accuracy, loss, and training speed increments differed according to the architecture. The latewood dataset showed the highest accuracy. The epochs and augmented datasets also positively affected accuracy, whereas the total part and non-augmented datasets had a negative effect on accuracy. Additionally, the augmented dataset tended to derive stable results and reached a convergence point earlier. In the present study, an augmented latewood dataset was the most important factor affecting classification performance and should be used for training CNNs.

**Keywords:** softwood; identification; convolutional neural networks (CNNs); VGG16; ResNet50; GoogLeNet

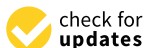



## 1. Introduction

Globally, there is excess demand for wood [1], resulting in a gradual appreciation of the value of wood as a resource in various sectors. Species identification is increasingly recognized as a significant process for enhancing the diversity and value of wood resources. This is required in numerous fields such as optimizing the utilization of conventional wood resources, protecting endangered species, and facilitating practical customs clearance operations.

The commonly employed traditional microscope-based identification method for wood species requires substantial time, cost, specialized training for equipment operation, and background knowledge cultivation to become an expert [2]. Furthermore, the traditional method has limitations such as potential variations in species classification among researchers and the need for information on wood species [3].

Research on the application of artificial neural networks to identify wood species is actively underway to simplify the species identification process and increase accessibility. Since 2010, the ImageNet Large-Scale Visual Recognition Challenge (ILSVRC), hosted by ImageNet (https://www.image-net.org (accessed on 10 April 2023)), has been conducted

worldwide to classify images using artificial neural networks. In particular, the ResNet model, the winner of ILSVRC 2015, achieved a performance of 3.6%, surpassing the 5.1% cognitive error rate known to occur in humans [4]. Since then, many artificial neural network models have achieved a 2%–5% cognitive error rate, thereby achieving remarkable technological advancements.

Deep learning has been used to reconstruct high-resolution images from low-quality images [5] and has improved high-resolution microscopy [6]. In the field of wood science, various attempts have been made to automate species identification, owing to the discovery of the potential of computer-based wood identification by the IAWA [7]. Machine vision has been applied by many researchers to identify wood samples, including species identification using convolutional neural networks and microscopic images [8], species analysis through thermal conductivity analysis by learning the thermal conductivity trend of each species using artificial neural networks [9], defect detection on wood surfaces using the Mask R-CNN model [10], object detection on wood surfaces for defect detection using the YOLO v3 algorithm [11], and detection of ray tissue features through pixel segmentation [12]. With advances in information technology, automated wood species identification has gradually become more sophisticated and precise.

Numerous studies have verified the performance of convolutional neural networks (CNNs) in wood species identification. For example, five Korean softwood species were successfully classified with high accuracy using an ensemble learning approach based on the CNN-based LeNet3 architecture [13]. Lopes et al. [14] also succeeded in classifying North American hardwood species from macroscale cross-sectional photographs of 10 species using a smartphone and portable magnifier. Fabijanska et al. [15] classified 14 softwood and hardwood species based on an R-CNN architecture using macroscale cross-sectional photographs of wood as a dataset. However, these studies aimed to classify wood species without discussing the factors that affect wood classification.

Therefore, in the present study, we selected VGG16 [16], ResNet50 [17], and GoogLeNet [18], which have demonstrated outstanding performance in the ImageNet Large-Scale Visual Recognition Challenge (ILSVRC), and basic CNN architecture. We verified their ability to classify 10 softwood species. Additionally, we analyzed the effects of various factors, such as the body part of the dataset, scale, data augmentation, and training iteration, on the accuracy and loss of the neural network models.

## 2. Materials and Methods

### 2.1. Materials

Ten softwood species were used as samples to construct a training dataset for the neural networks. Six commercially imported species were purchased from the market, and four domestic species were obtained from the research forest of Kangwon National University. Detailed information on the wood samples is provided in Table 1.

**Table 1.** Sample information.

| Common Name | Scientific Name | Origin | Supplier |
|---|---|---|---|
| Cedar | *Cryptomeria japonica* | Japan | |
| Japanese cypress | *Chamaecyparis obtusa* | Japan | |
| Mugo pine | *Pinus mugo* | Finland | W Wood Co., Ltd. |
| Radiata pine | *Pinus radiata* | USA | (Daejeon, Republic of Korea) |
| Spruce | *Picea abies* | Estonia | |
| Yin shan shu | *Cathaya argyrophylla* | Russia | |
| Korean red pine | *Pinus densiflora* | | Research forest of |
| Korean white pine | *Pinus koraiensis* | Chuncheon, | Kangwon National University |
| Metasequoia | *Metasequoia glyptostroboides* | Republic of Korea | (Chuncheon, Republic of Korea: |
| Juniper | *Juniperus chinensis* | | 37.7748857, 127.8134654) |

## 2.2. Methods

### 2.2.1. Sample Preparation for the Dataset

We employed common microscopy for wood anatomy [19,20]. The four species harvested from the heartwood and sapwood at Kangwon National University were processed into small blocks with dimensions of 30 (R) mm × 10 (T) mm × 10 (L) mm from heartwood and sapwood, whereas the six other species purchased as wood panels were randomly processed into small blocks. We collected 15–20 small blocks from each species. These blocks were converted to 20–30 μm-thick slices using a sliding microtome (MSL Model; Nippon Optical Works, Nagano, Japan). The sections were stained with a 1% safranin solution, dehydrated using an ethanol series, cleared with xylene, and mounted on a permanent slide using Canada balsam.

To collect the dataset, cross-sections were observed with an Infinity-1 camera (1.3 MP; Lumenera, Ottawa, ON, Canada) connected to an ECLIPSE E600 optical microscope (NIKON, Tokyo, Japan), with 4× and 20× objective lenses. Micrographs were taken and analyzed using the i-Solution Lite image analysis software (IMT, Victoria, BC, Canada).

To analyze the effect of the micrograph capture location on the performance of the artificial neural networks, three types of dataset were prepared: total part, earlywood, and latewood datasets. The entire dataset comprised a total part of earlywood and latewood captured with a 4× objective lens. The earlywood and latewood datasets were captured using a 20× objective lens. The actual areas captured in the micrographs under the 4× and 20× objective lenses were approximately 10.89 mm² and 0.44 mm², respectively.

### 2.2.2. Dataset Preprocessing

Prior to training, the RGB coefficients were reduced using a 1/255 ratio to decrease the range to 0–1. The ImageDataGenerator function was used to augment the dataset and to evaluate its impact. Various argument factors were applied to augment the dataset, such as a rotation range of 10°, width_shift_range and height_shift_range of 10%, zoom_range of 20%, and horizontal_flip and vertical_flip. An example of dataset augmentation is shown in Figure 1.

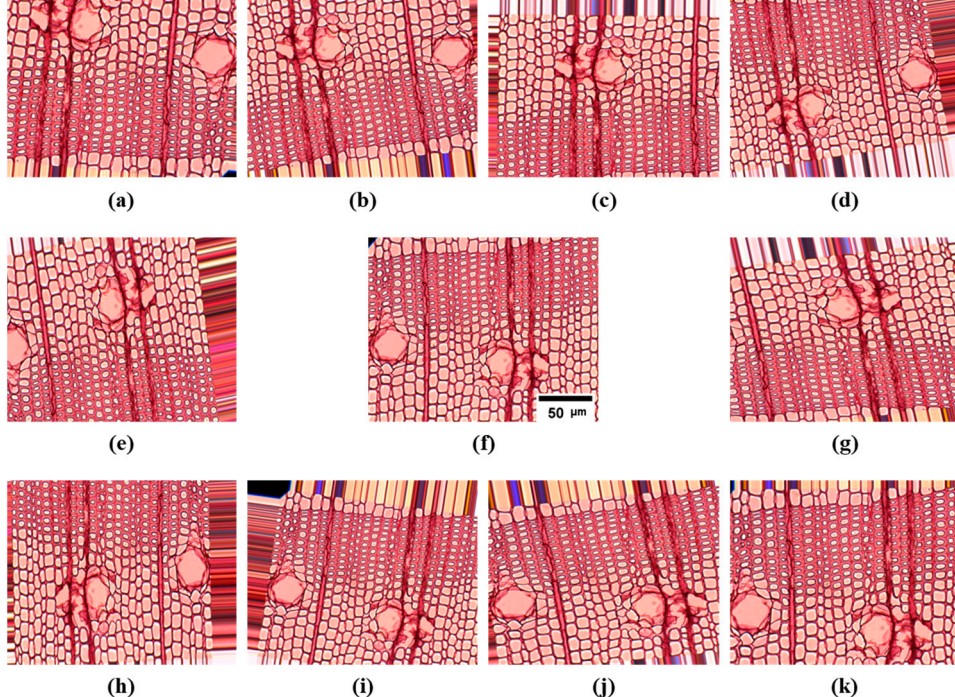

**Figure 1.** Example micrographs of image augmentation: (**a**) original image; (**b**–**k**) augmented images.

We used 80% and 20% of the prepared dataset for training and testing, respectively, based on the size of the non-augmented dataset. Data augmentation was applied only to the training dataset and not to the testing dataset. Based on these criteria, the quantities of the datasets utilized in the present study are listed in Table 2 for the non-augmented state and in Table 3 for the augmented state.

**Table 2.** Composition of training and test dataset with non-augmentation.

| Common Name | Scientific Name | 40× Dataset (Total) | | | 200× Dataset (Earlywood, Latewood) | | |
|---|---|---|---|---|---|---|---|
| | | Train | Test | Sum | Train | Test | Sum |
| Cedar | *Cryptomeria japonica* | 160 | 40 | 200 | 160 | 40 | 200 |
| Japanese cypress | *Chamaecyparis obtusa* | 160 | 40 | 200 | 160 | 40 | 200 |
| Mugo pine | *Pinus mugo* | 160 | 40 | 200 | 160 | 40 | 200 |
| Radiata pine | *Pinus radiata* | 160 | 40 | 200 | 160 | 40 | 200 |
| Spruce | *Picea abies* | 160 | 40 | 200 | 160 | 40 | 200 |
| Yin shan shu | *Cathaya argyrophylla* | 160 | 40 | 200 | 160 | 40 | 200 |
| Korean red pine | *Pinus densiflora* | 160 | 40 | 200 | 160 | 40 | 200 |
| Korean white pine | *Pinus koraiensis* | 160 | 40 | 200 | 160 | 40 | 200 |
| Metasequoia | *Metasequoia glyptostroboides* | 160 | 40 | 200 | 160 | 40 | 200 |
| Juniper | *Juniperus chinensis* | 160 | 40 | 200 | 160 | 40 | 200 |
| Sum | | 1600 | 400 | 2000 | 1600 | 400 | 2000 |

**Table 3.** Composition of the training and test datasets with augmentation.

| Common Name | Scientific Name | 40× Dataset | | | 200× Dataset | | | | | |
|---|---|---|---|---|---|---|---|---|---|---|
| | | | | | Earlywood | | | Latewood | | |
| | | Train | Test | Sum | Train | Test | Sum | Train | Test | Sum |
| Cedar | *Cryptomeria japonica* | 160 | 40 | 200 | 1764 | 40 | 1804 | 1768 | 40 | 1808 |
| Japanese cypress | *Chamaecyparis obtusa* | 160 | 40 | 200 | 1774 | 40 | 1814 | 1754 | 40 | 1794 |
| Mugo pine | *Pinus mugo* | 160 | 40 | 200 | 1775 | 40 | 1815 | 1772 | 40 | 1812 |
| Radiata pine | *Pinus radiata* | 160 | 40 | 200 | 1781 | 40 | 1821 | 1775 | 40 | 1815 |
| Spruce | *Picea abies* | 160 | 40 | 200 | 1773 | 40 | 1813 | 1781 | 40 | 1821 |
| Yin shan shu | *Cathaya argyrophylla* | 160 | 40 | 200 | 1762 | 40 | 1802 | 1776 | 40 | 1816 |
| Korean red pine | *Pinus densiflora* | 160 | 40 | 200 | 1783 | 40 | 1823 | 1785 | 40 | 1825 |
| Korean white pine | *Pinus koraiensis* | 160 | 40 | 200 | 1767 | 40 | 1807 | 1777 | 40 | 1817 |
| Metasequoia | *Metasequoia glyptostroboides* | 160 | 40 | 200 | 1764 | 40 | 1804 | 1779 | 40 | 1819 |
| Juniper | *Juniperus chinensis* | 160 | 40 | 200 | 1784 | 40 | 1822 | 1768 | 40 | 1808 |
| Sum | | 1600 | 400 | 2000 | 17,739 | 400 | 18,125 | 17,735 | 400 | 18,135 |

Micrographs of the dataset were constructed with 1280 × 1024 pixels (1,310,720 pixels) and resized to 224 × 224 pixels (50,176 pixels) to conserve the system resources used during training.

### 2.2.3. Verification Factors Influencing Neural Networks

The classification performance and factors influencing the performance were analyzed using four models, VGG16, ResNet, and GoogLeNet, which had excellent performance in ILSVRC, and a basic CNN architecture consisting of 12 layers. Their structural features were described in Table 4, and the basic CNN architecture was schematized in Figure 2.

**Table 4.** Features of each neural network.

| Architecture | Layers | Convolutional Filter | Structural Features |
|---|---|---|---|
| VGG16 | 25 | $3 \times 3$ convolutional layer | - 13 convolutional layers<br>- 3 dense layers |
| ResNet50 | 50 | $3 \times 3$ convolutional layer | - Introducing the concept of skip connection to solve the gradient vanishing problem<br>- Shortcut connection between inputs and outputs in a convolutional layer |
| GoogLeNet | 16 | $1 \times 1$ convolutional layer<br>$3 \times 3$ convolutional layer<br>$5 \times 5$ convolutional layer<br>$3 \times 3$ pooling layer | - Repetition of inception module<br>- Parallel performance of $1 \times 1$, $3 \times 3$, and $5 \times 5$ convolutional operations |
| Basic CNN | 12 | $3 \times 3$ convolutional layer | - 4 convolutional layers<br>- 4 max pooling layers<br>- 2 dense layers<br>- 1 flatten layer<br>- 1 dropout layer |

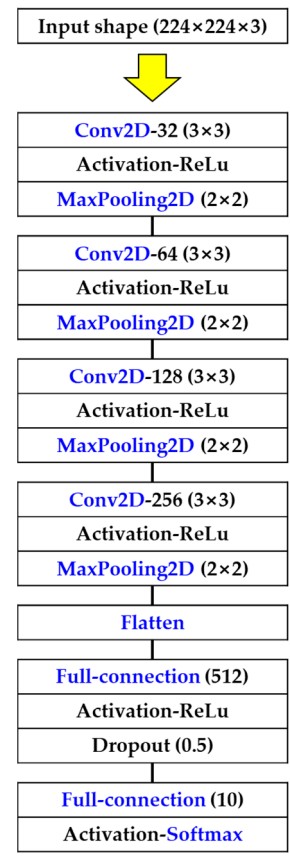

**Figure 2.** Basic CNN architecture in the present study.

The four neural network models used in this study are based on the reports of Simonyan and Zisserman [16], He et al. [17], Szegedy et al. [18], and Elgendy [21], which describe the characteristics of each model.

To classify the 10 species using the results generated by the artificial neural networks, the softmax function was used as the activation function. For compilation, a categorical cross-entropy function was applied as a multi-classification loss function. For the optimization, the RMSprop function was applied to the VGG16 model, whereas the SGD function was used for the ResNet50 and GoogLeNet models.

### 2.2.4. Correlation Analysis between Factors

In this study, the Pearson correlation coefficients among the variables were analyzed using bivariate correlation analysis in SPSS 26.0 (IBM, New York, NY, USA). Nominal variables, including the dataset collection region (total, earlywood, and latewood) and augmentation for analysis, were applied, while accuracy and loss were applied as scale variables. Furthermore, we analyzed homogeneous subsets among the results using Duncan's post-hoc analysis with one-way ANOVA.

## 3. Results and Discussion

### 3.1. VGG16 Architecture

Figure 3 shows the results of the wood species classification using the VGG16 model. Under all conditions, as the number of epochs increased, the loss decreased, and the classification accuracy increased. In terms of the composition of the dataset, both the augmented and non-augmented datasets, in terms of stability, showed the best performance in the latewood dataset, followed by the earlywood dataset and the total part dataset. However, the use of augmented datasets resulted in faster stabilization of the results compared to non-augmented datasets. The test dataset showed greater instability than the training dataset, whereas the classification accuracy increased proportionally with the number of epochs, showing results similar to those of the training dataset. Therefore, to achieve more stable performance using the VGG16 architecture to classify wood species, the dataset should be constructed with a latewood part. Training and testing processes should be conducted after expanding the dataset through augmentation to achieve efficient performance.

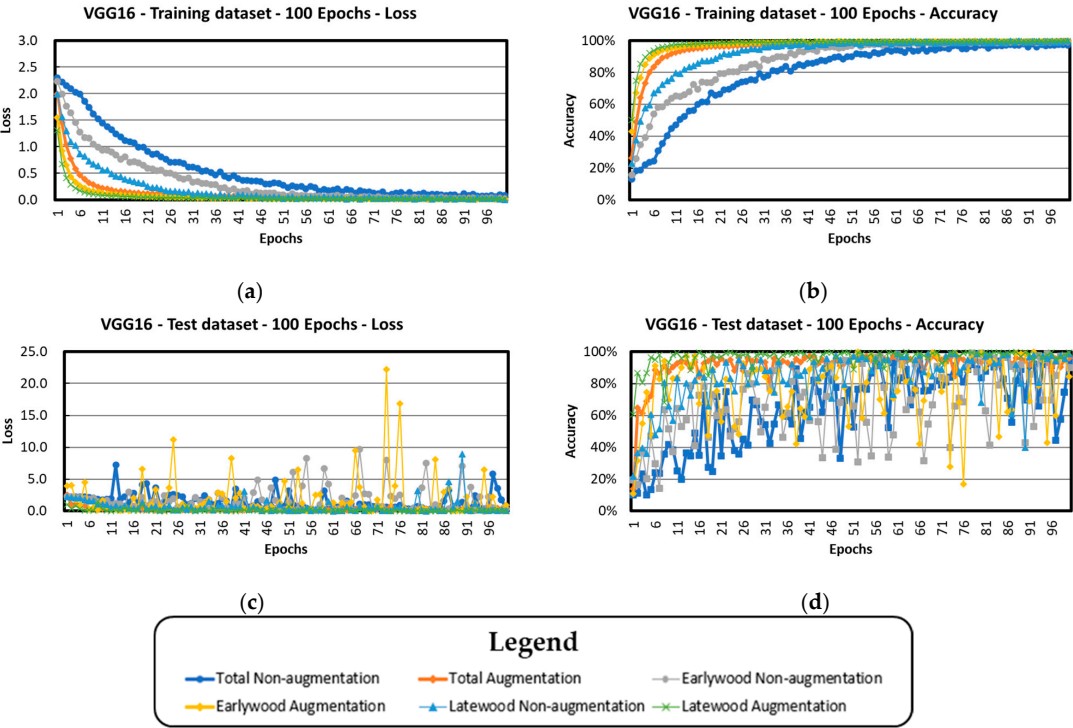

**Figure 3.** Verification results of VGG16 architecture: loss (**a**) and accuracy (**b**) variation in the training dataset with increasing epochs. Loss (**c**) and accuracy (**d**) variation in the test dataset with increasing epochs.

Table 5 presents the results of the homogeneous subsets among the verification conditions of the dataset based on the verification results of the VGG16 architecture, as shown in Figure 3. During the training process, the augmented dataset showed a comparable trend in both loss and accuracy across all parts, including the total, earlywood, and latewood parts, whereas the non-augmented dataset was classified as a self-reliant group among the parts. During the test process, the augmented datasets of the total and latewood parts showed similar trends in both loss and accuracy, whereas the earlywood parts showed similar accuracy and loss in the non-augmented and augmented datasets.

**Table 5.** Homogeneous subset output of the VGG16 model.

| Dataset | | Total (40×) | | Earlywood (200×) | | Latewood (200×) | |
|---|---|---|---|---|---|---|---|
| | | NAug | Aug | NAug | Aug | NAug | Aug |
| Train dataset | Loss | 0.546 [d] | 0.136 [ab] | 0.340 [c] | 0.080 [a] | 0.200 [b] | 0.058 [a] |
| | accuracy | 0.803 [a] | 0.955 [cd] | 0.876 [b] | 0.972 [d] | 0.926 [c] | 0.980 [d] |
| Test dataset | Loss | 1.500 [c] | 0.364 [ab] | 1.769 [c] | 1.968 [c] | 0.721 [b] | 0.157 [a] |
| | accuracy | 0.649 [a] | 0.916 [d] | 0.682 [a] | 0.780 [b] | 0.844 [c] | 0.963 [d] |

Note: The same superscript lowercase letters beside the mean values in the same row denote non-significant outcomes at the 5% significance level for comparison between datasets. NAug: non-augmented; Aug: augmented.

*3.2. ResNet50 Architecture*

Figure 4 shows the results of the wood species classification using the ResNet50 architecture. Under all conditions, an increase in epochs led to a decrease in the loss and an increase in the classification accuracy. The difference in stabilization due to the composition and augmentation of the dataset was not clearly analyzed, but most datasets under all conditions reached a stabilization state at approximately 10–15 epochs during the training process. However, the total, 40× image, and non-augmented condition datasets had the longest stabilization time: that is, approximately 20 epochs.

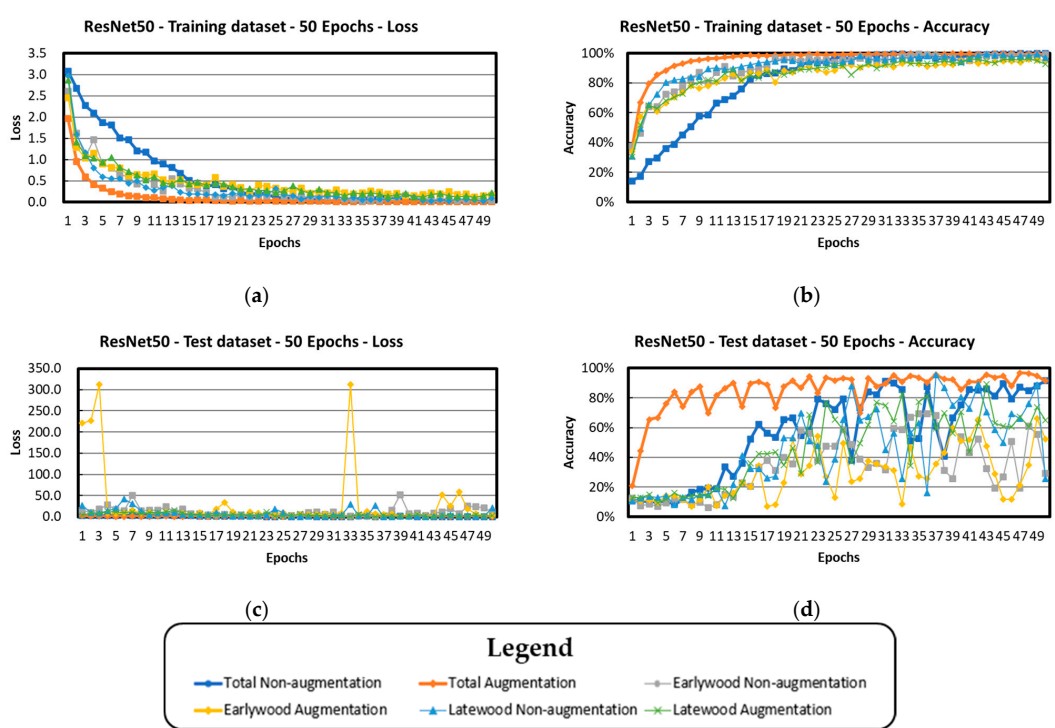

**Figure 4.** Verification results of ResNet50 architecture. Loss (**a**) and accuracy (**b**) variation in the training dataset with increasing epochs. Loss (**c**) and accuracy (**d**) variation in the test dataset with increasing epochs.

The test dataset showed a tendency for the loss to decrease and the accuracy to increase with increasing epochs, but it exhibited much greater instability than the training dataset. This instability can be attributed to a shortage of the test dataset and the occurrence of gradient vanishing owing to the deep neural network structure of the ResNet architecture, resulting in greater fluctuations in accuracy and loss.

Table 6 presents the results of the analysis of the homogeneous subsets of the dataset for each verification condition based on the results of the ResNet50 architecture, as shown in Figure 4. The datasets used for the ResNet50 architecture belong to the same group as the augmented datasets under all conditions. The non-augmented dataset was classified into the same subset as the earlywood and latewood datasets during the training process. Only the accuracies using the total part and earlywood datasets were classified as the same subset during the test process.

**Table 6.** Homogeneous subset analysis of the ResNet50 model.

| Dataset | | Total (40×) | | Earlywood (200×) | | Latewood (200×) | |
|---|---|---|---|---|---|---|---|
| | | NAug | Aug | NAug | Aug | NAug | Aug |
| Train dataset | Loss | 0.544 c | 0.121 ab | 0.287 b | 0.064 a | 0.237 ab | 0.048 a |
| | accuracy | 0.822 a | 0.959 bc | 0.900 b | 0.978 c | 0.923 bc | 0.984 c |
| Test dataset | Loss | 1.697 c | 0.637 ab | 2.435 d | 0.334 a | 1.100 b | 0.090 a |
| | accuracy | 0.572 a | 0.854 c | 0.578 a | 0.927 cd | 0.765 b | 0.974 d |

Note: The same superscript lowercase letters beside the mean values in the same row denote non-significant outcomes at the 5% significance level for comparison between datasets. NAug: non-augmented; Aug: augmented.

### 3.3. GoogLeNet Architecture

Figure 5 shows the results of wood species classification using the GoogLeNet (Inception v1) model. Under all conditions, the loss decreased and the classification accuracy increased as the number of epochs increased; however, the variations in loss and accuracy were more gradual compared with the verification results of other neural network architectures. The influence of the collected dataset was more significant than that of data augmentation, and both the loss and accuracy were highest for the latewood dataset, followed by the earlywood dataset and the total dataset. The GoogLeNet architecture showed a relatively delayed stabilization of the loss and accuracy compared to the other architectures analyzed in the present study. The latewood dataset showed the fastest stabilization achieved in the range of approximately 160–170 epochs. In contrast, the loss and accuracy did not stabilize until the end of learning at 200 epochs in the earlywood dataset.

Table 7 presents an analysis of the homogeneous subsets of the datasets across the verification conditions based on the results of the GoogLeNet architecture, as shown in Figure 5. Most of the datasets used for the GoogLeNet architecture were classified into significantly different groups, regardless of augmentation and composition. However, the earlywood test dataset exhibited the same subset only under both non-augmented and augmented conditions.

**Table 7.** Homogeneous subset output of the GoogLeNet model.

| Dataset | | Total (40×) | | Earlywood (200×) | | Latewood (200×) | |
|---|---|---|---|---|---|---|---|
| | | NAug | Aug | NAug | Aug | NAug | Aug |
| Train dataset | Loss | 1.429 e | 1.680 f | 1.113 c | 1.246 d | 0.711 a | 0.972 b |
| | accuracy | 0.480 b | 0.387 a | 0.595 d | 0.549 c | 0.734 e | 0.639 d |
| Test dataset | Loss | 1.462 d | 1.609 e | 1.138 c | 1.159 c | 0.659 a | 0.898 b |
| | accuracy | 0.464 b | 0.411 a | 0.582 c | 0.575 c | 0.752 e | 0.673 d |

Note: The same superscript lowercase letters beside the mean values in the same row denote non-significant outcomes at the 5% significance level for comparison between datasets. NAug: non-augmented; Aug: augmented.

### 3.4. Basic CNN Architecture

Figure 6 shows the results of wood species classification using a basic CNN architecture. As the number of epochs increased in both the training and test datasets, the loss decreased and the accuracy increased until 100 epochs of the last training. The accuracy and loss of the verification results were influenced by the augmentation and composition of the datasets. The augmented datasets showed rapid training and stabilization within the range of 10–20 epochs, whereas the non-augmented datasets progressed slowly during training and stabilization up to 40–80 epochs. Depending on the composition of the datasets, the training and stabilization speeds of the latewood dataset were the best, followed by those of the earlywood and total datasets.

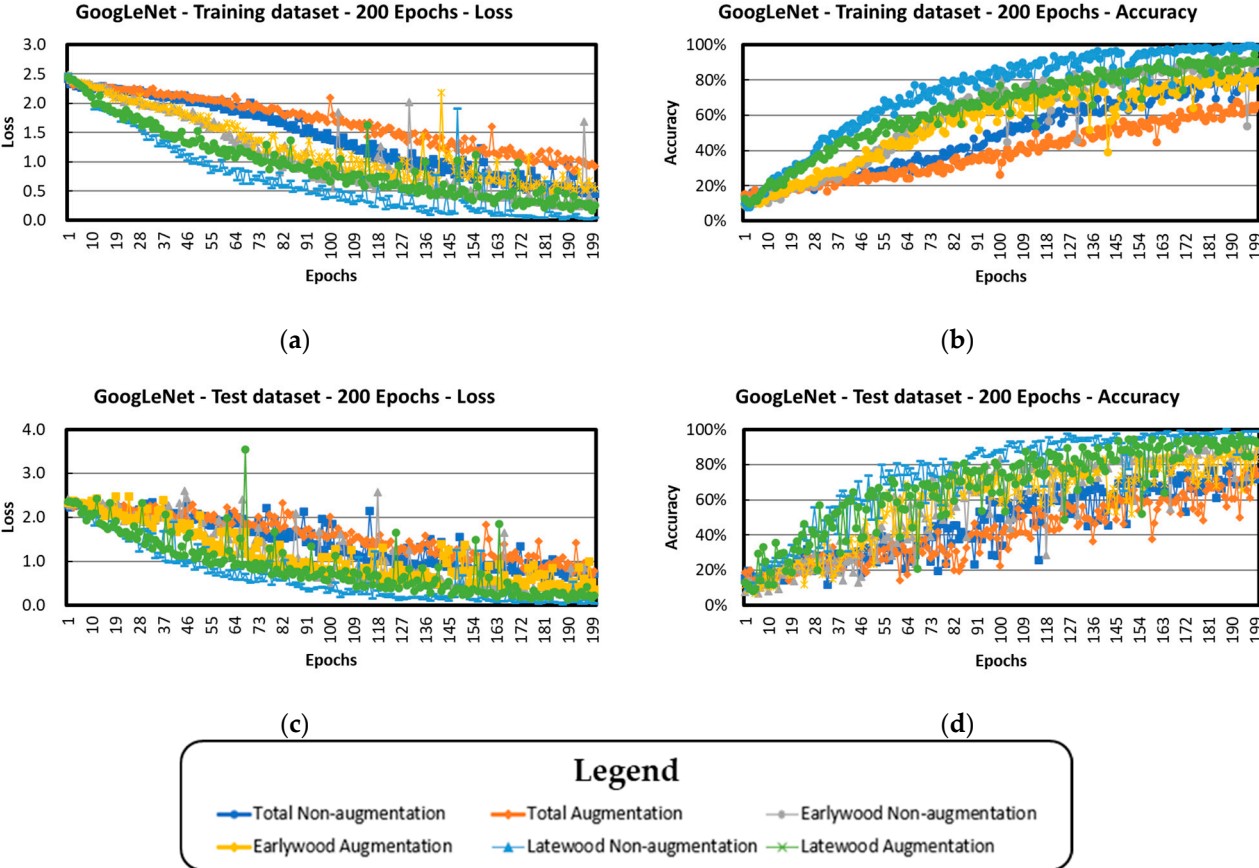

(a)　　　　　　　　　　　　　　　　(b)

(c)　　　　　　　　　　　　　　　　(d)

**Figure 5.** Verification results of GoogLeNet architecture. Loss (**a**) and accuracy (**b**) variation in the training dataset with increasing epochs. Loss (**c**) and accuracy (**d**) variation in the test dataset with increasing epochs.

Table 8 presents the results of the analysis of the homogeneous subsets for each verification condition of the basic CNN architecture, as shown in Figure 6. The augmented datasets used in the basic CNN architecture were classified into the same group as those in the training condition. However, in the non-aggregated datasets, all conditions were independent of the training and test processes. During the test process, only the augmented earlywood and latewood datasets were in the same group, whereas the other conditions were independent groups.

**Table 8.** Homogeneous subset output of the basic CNN model.

| Dataset | | Total (40×) | | Earlywood (200×) | | Latewood (200×) | |
|---|---|---|---|---|---|---|---|
| | | NAug | Aug | NAug | Aug | NAug | Aug |
| Train dataset | Loss | 0.925 [d] | 0.205 [ab] | 0.671 [c] | 0.130 [a] | 0.349 [b] | 0.101 [a] |
| | accuracy | 0.675 [a] | 0.930 [d] | 0.777 [b] | 0.957 [d] | 0.873 [c] | 0.965 [d] |
| Test dataset | Loss | 1.038 [e] | 0.466 [c] | 0.832 [d] | 0.238 [b] | 0.503 [c] | 0.074 [a] |
| | accuracy | 0.642 [a] | 0.873 [d] | 0.714 [b] | 0.927 [e] | 0.819 [c] | 0.972 [e] |

Note: The same superscript lowercase letters beside the mean values in the same row denote non-significant outcomes at the 5% significance level for comparison between datasets. NAug: non-augmented; Aug: augmented.

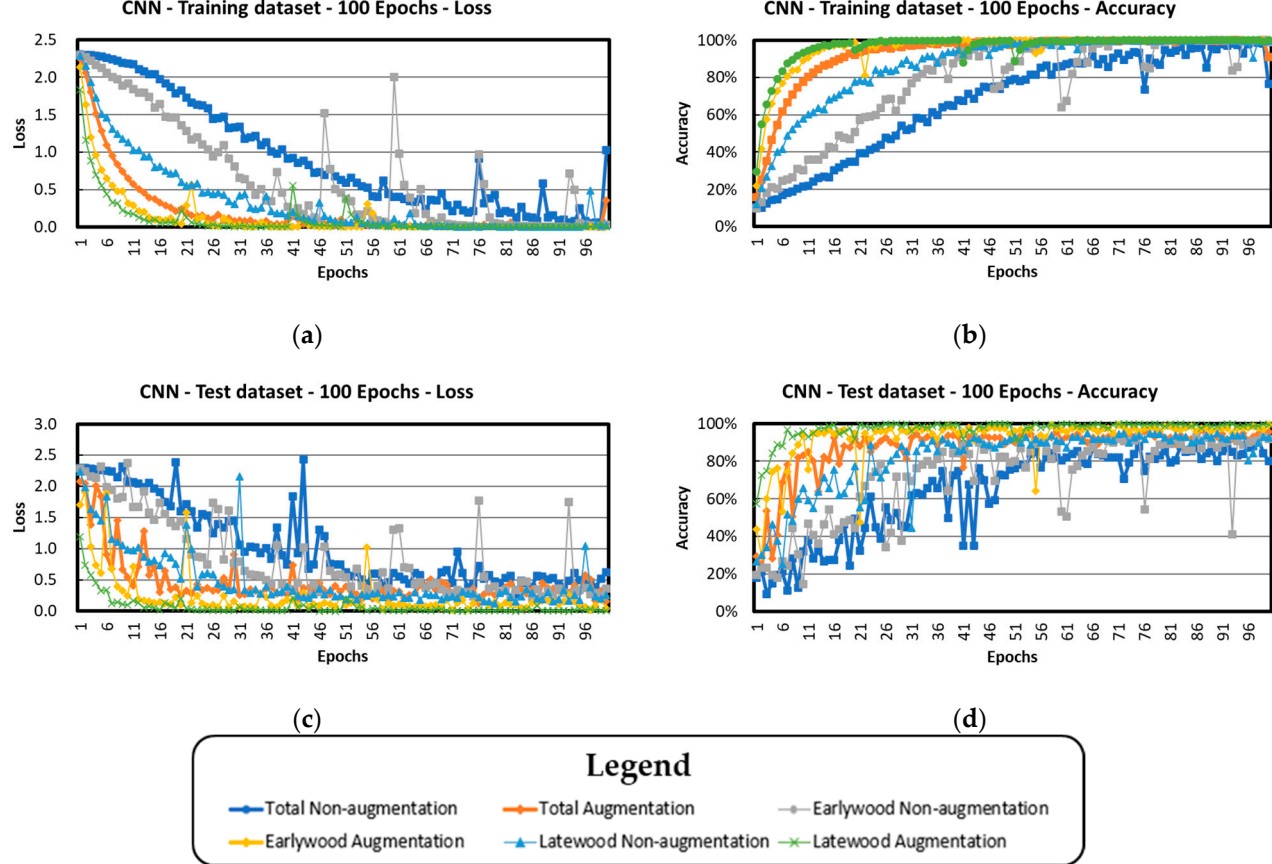

(**a**)

(**b**)

(**c**)

(**d**)

**Figure 6.** Verification results of basic CNN architecture. Loss (**a**) and accuracy (**b**) variation in the training dataset with increasing epochs. Loss (**c**) and accuracy (**d**) variation in the test dataset with increasing epochs.

### 3.5. General Trend

Table 9 presents the comparison of the average accuracy at the final five stages during the testing process for the four architectures analyzed in the present study. The results show that the average accuracy at the final five stages varies depending on the collected part of the dataset and the data augmentation. Regarding the influence of the collected part of the dataset on accuracy, the highest accuracy was achieved in the latewood dataset for training, followed by the earlywood dataset, whereas the total part dataset exhibited the lowest accuracy. Data augmentation demonstrated an improvement in classification accuracy with employing augmented datasets for all collected parts, resulting in higher classification accuracy compared to non-augmented datasets.

**Table 9.** Average accuracy of the last five steps by neural network architecture (unit: %).

|  | Total (40×) | | Earlywood (200×) | | Latewood (200×) | |
|---|---|---|---|---|---|---|
|  | **NAug** | **Aug** | **NAug** | **Aug** | **NAug** | **Aug** |
| VGG16 | 72.5 (0.3) | 92.9 (0.1) | 87.9 (0.6) | 86.8 (0.7) | 96.6 (0.9) | 97.8 (1.4) |
| ResNet50 | 86.3 (4.5) | 93.6 (3.8) | 91.9 (18.0) | 99.1 (22.2) | 96.1 (23.8) | 99.5 (5.5) |
| GoogLeNet | 73.9 (2.7) | 70.5 (6.8) | 83.8 (6.1) | 81.5 (7.9) | 98.1 (1.4) | 91.6 (4.1) |
| CNN | 86.3 (4.3) | 94.0 (1.9) | 91.4 (1.2) | 98.3 (0.9) | 88.1 (5.4) | 99.3 (0.5) |

Note: Numbers in parentheses are standard deviations. NAug: non-augmented; Aug: augmented.

Table 10 shows the correlation of the variables applied to the training and test process of the four architectures, namely, VGG16, ResNet50, GoogLeNet, and basic CNN, which were used to classify the 10 softwood species in the present study.

**Table 10.** Correlation of the factors influencing convolutional neural networks.

| N = 2700 | Epochs | Loss (Train) | Accuracy (Train) | Loss (Test) | Accuracy (Test) | Position (Total) | Position (Earlywood) | Position (Latewood) | Augmentation (No) | Augmentation (Yes) |
|---|---|---|---|---|---|---|---|---|---|---|
| Epochs | 1 | −0.157 ** $p = 0.000$ | 0.166 ** $p = 0.000$ | −0.225 ** $p = 0.000$ | 0.267 ** $p = 0.000$ | 0.000 $p = 0.000$ | 0.000 $p = 0.000$ | 0.000 $p = 0.000$ | 0.000 $p = 0.000$ | 0.000 $p = 0.000$ |
| Loss (train) | −0.157 ** $p = 0.000$ | 1 | −0.996 ** $p = 0.000$ | 0.442 ** $p = 0.000$ | −0.886 ** $p = 0.000$ | 0.227 ** $p = 0.000$ | −0.012 $p = 0.521$ | −0.214 ** $p = 0.000$ | 0.073 ** $p = 0.000$ | −0.073 ** $p = 0.000$ |
| Accuracy (train) | 0.166 ** $p = 0.000$ | −0.996 ** $p = 0.000$ | 1 | −0.442 ** $p = 0.000$ | 0.889 ** $p = 0.000$ | −0.219 ** $p = 0.000$ | 0.018 $p = 0.337$ | 0.201 ** $p = 0.000$ | −0.068 ** $p = 0.000$ | 0.068 ** $p = 0.000$ |
| Loss (test) | −0.225 ** $p = 0.000$ | 0.442 ** $p = 0.000$ | −0.442 ** $p = 0.000$ | 1 | −0.719 ** $p = 0.000$ | 0.120 ** $p = 0.000$ | 0.127 ** $p = 0.000$ | −0.248 ** $p = 0.000$ | 0.137 ** $p = 0.000$ | −0.137 ** $p = 0.000$ |
| Accuracy (test) | 0.267 ** $p = 0.000$ | −0.886 ** $p = 0.000$ | 0.889 ** $p = 0.000$ | −0.719 ** $p = 0.000$ | 1 | −0.238 ** $p = 0.000$ | −0.051 ** $p = 0.008$ | 0.289 ** $p = 0.000$ | −0.172 ** $p = 0.000$ | 0.172 ** $p = 0.000$ |
| Position (total) | 0.000 $p = 0.000$ | 0.227 ** $p = 0.000$ | −0.219 ** $p = 0.000$ | 0.120 ** $p = 0.000$ | −0.238 ** $p = 0.000$ | 1 | −0.500 ** $p = 0.000$ | −0.500 ** $p = 0.000$ | 0.000 $p = 0.000$ | 0.000 $p = 0.000$ |
| Position (earlywood) | 0.000 $p = 0.000$ | −0.012 $p = 0.521$ | 0.018 $p = 0.337$ | 0.127 ** $p = 0.000$ | −0.051 ** $p = 0.008$ | −0.500 ** $p = 0.000$ | 1 | −0.500 ** $p = 0.000$ | 0.000 $p = 0.000$ | 0.000 $p = 0.000$ |
| Position (latewood) | 0.000 $p = 0.000$ | −0.214 ** $p = 0.000$ | 0.201 ** $p = 0.000$ | −0.248 ** $p = 0.000$ | 0.289 ** $p = 0.000$ | −0.500 ** $p = 0.000$ | −0.500 ** $p = 0.000$ | 1 | 0.000 $p = 0.000$ | 0.000 $p = 0.000$ |
| Augmentation (yes) | 0.000 $p = 0.000$ | 0.073 ** $p = 0.000$ | −0.068 ** $p = 0.000$ | 0.137 ** $p = 0.000$ | −0.172 ** $p = 0.000$ | 0.000 $p = 0.000$ | 0.000 $p = 0.000$ | 0.000 $p = 0.000$ | 1 | −0.000 ** $p = 0.000$ |
| Augmentation (no) | 0.000 $p = 0.000$ | −0.073 ** $p = 0.000$ | 0.068 ** $p = 0.000$ | −0.137 ** $p = 0.000$ | 0.172 ** $p = 0.000$ | 0.000 $p = 0.000$ | 0.000 $p = 0.000$ | 0.000 $p = 0.000$ | −0.000 ** $p = 0.000$ | 1 |

** The correlation is significant at the 0.01 level (two-tailed).

The loss tended to decrease with increasing epochs in both the training and test processes, whereas the accuracy tended to increase in proportion to the epochs. This tendency was more significant in the test dataset than in the training dataset, which could be attributed to the performance improvement of the test dataset due to weight updates through training. In addition, accuracy tended to decrease with the application of the total part or the non-augmented dataset, whereas it increased with the application of the latewood part or the augmented dataset. The loss tended to be opposite to that of the accuracy.

Table 11 presents the correlation between variables within the final five epochs during the training and test processes for the four architectures. Due to biased data, it shows partially different results from Table 10, which compared the overall correlation of variables across all epochs. Representative differences include increasing loss and decreasing accuracy as the epochs progress during the training process, as well as the decreasing trend of accuracy with increasing epochs during the test process. The influence of the collected part of the dataset and augmentation was consistent with the trends observed in Table 10.

**Table 11.** Correlation of the factors within the last five steps influencing convolutional neural networks.

| N = 120 | Epochs | Loss (Train) | Accuracy (Train) | Loss (Test) | Accuracy (Test) | Position (Total) | Position (Earlywood) | Position (Latewood) | Augmentation (No) | Augmentation (Yes) |
|---|---|---|---|---|---|---|---|---|---|---|
| Epochs | 1 | 0.658 ** $p = 0.000$ | −0.703 ** $p = 0.000$ | 0.107 $p = 0.246$ | −0.390 ** $p = 0.000$ | 0.000 $p = 1.000$ | 0.000 $p = 1.000$ | 0.000 $p = 1.000$ | 0.000 $p = 1.000$ | 0.000 $p = 1.000$ |
| Loss (train) | 0.658 ** $p = 0.000$ | 1 | −0.982 ** $p = 0.000$ | 0.181 * $p = 0.047$ | −0.568 ** $p = 0.000$ | 0.212 * $p = 0.020$ | 0.026 $p = 0.782$ | −0.238 ** $p = 0.009$ | −0.023 $p = 0.799$ | 0.023 $p = 0.799$ |
| Accuracy (train) | −0.703 ** $p = 0.000$ | −0.982 ** $p = 0.000$ | 1 | −0.189 * $p = 0.039$ | 0.591 ** $p = 0.000$ | −0.230 * $p = 0.012$ | −0.021 $p = 0.821$ | 0.251 ** $p = 0.006$ | 0.068 $p = 0.463$ | −0.068 $p = 0.463$ |
| Loss (test) | 0.107 $p = 0.246$ | 0.181 * $p = 0.047$ | −0.189 * $p = 0.039$ | 1 | −0.827 ** $p = 0.000$ | 0.328 ** $p = 0.000$ | −0.033 $p = 0.721$ | −0.295 ** $p = 0.001$ | 0.165 $p = 0.071$ | −0.165 $p = 0.071$ |
| Accuracy (test) | −0.390 ** $p = 0.000$ | −0.568 ** $p = 0.000$ | 0.591 ** $p = 0.000$ | −0.827 ** $p = 0.000$ | 1 | −0.420 ** $p = 0.000$ | 0.012 $p = 0.896$ | 0.408 ** $p = 0.000$ | −0.209 * $p = 0.022$ | 0.209 * $p = 0.022$ |
| Position (total) | 0.000 $p = 1.000$ | 0.212 * $p = 0.020$ | −0.230 * $p = 0.012$ | 0.328 ** $p = 0.000$ | −0.420 ** $p = 0.000$ | 1 | −0.500 ** $p = 0.000$ | −0.500 ** $p = 0.000$ | 0.000 $p = 1.000$ | 0.000 $p = 1.000$ |
| Position (earlywood) | 0.000 $p = 1.000$ | 0.026 $p = 0.782$ | −0.021 $p = 0.821$ | −0.033 $p = 0.721$ | 0.012 $p = 0.896$ | −0.500 ** $p = 0.000$ | 1 | −0.500 ** $p = 0.000$ | $p = 0.000$ | $p = 0.000$ |
| Position (latewood) | 0.000 $p = 1.000$ | −0.238 ** $p = 0.009$ | 0.251 ** $p = 0.006$ | −0.295 ** $p = 0.001$ | 0.408 ** $p = 0.000$ | −0.500 ** $p = 0.000$ | −0.500 ** $p = 0.000$ | 1 | 0.000 $p = 1.000$ | 0.000 $p = 1.000$ |
| Augmentation (yes) | 0.000 $p = 1.000$ | −0.023 $p = 0.799$ | 0.068 $p = 0.463$ | 0.165 $p = 0.071$ | −0.209 * $p = 0.022$ | 0.000 $p = 1.000$ | 0.000 $p = 1.000$ | 0.000 $p = 1.000$ | 1 | −1.000 ** $p = 0.000$ |
| Augmentation (no) | 0.000 $p = 1.000$ | 0.023 $p = 0.799$ | −0.068 $p = 0.463$ | −0.165 $p = 0.071$ | 0.209 * $p = 0.022$ | 0.000 $p = 1.000$ | 0.000 $p = 1.000$ | 0.000 $p = 1.000$ | −1.000 ** $p = 0.000$ | 1 |

** The correlation is significant at the 0.01 level (two-tailed). * The correlation is significant at the 0.05 level (two-tailed).

## 4. Discussion

In this study, we analyzed the classification performance and its influencing factors on the classification of softwood species in four architectures: VGG16, ResNet50, GoogLeNet, and basic CNN. The four architectures based on neural networks showed excellent classification performance of over 90% for wood species classification, showing similar performance as in previous studies [22–25]. In the architectures, the accuracy increased and the loss decreased with increasing epochs. This is due to the weight updates resulting from the epoch increment, because training in deep learning is the process of finding weights to minimize the loss function [26] and updating the internal parameters such as weights through training [27]. In the initial stages of training, the weights were assigned random values, resulting in low accuracy and a high loss; however, as the training progressed, the weights were adjusted toward the correct output [28], leading to a decrease in loss. Sufficient training repetition resulted in the derivation of weight values that minimized the loss of function [29]. Therefore, the results of the present study can be explained by the fact that the weights of the dataset consisted of micrographs that were appropriately adjusted with increasing neural network training.

The gradients of accuracy and loss curves observed during the training and testing processes were influenced by the size of the convolutional filters. Camgozlu and Kutlu [30] investigated the influence of the size of image and convolutional filter on deep learning and reported that smaller convolutional filters lead to improved accuracy and reduced training time. Ahmed and Karim [31] also analyzed the impact of convolutional filter size and quantity of convolutional filter on classification accuracy and reported that smaller filter sizes result in superior performance. In the present study, the four architectures used for classifying softwood species, VGG16, ResNet50, GoogLeNet, and basic CNN, can be divided into two types based on the size of the convolutional filters. Three architectures utilized 3 × 3 convolutional filters, VGG16, ResNet50, and basic CNN, exhibiting rapid stabilizations in the accuracy and loss curves during the early stages of training. In contrast, GoogLeNet consisted of three different sizes of convolutional filters (1 × 1, 3 × 3, and 5 × 5) within each inception module, resulting in a more gradual slope compared to other convolutional neural network architectures.

The analysis of factors influencing the performance of neural networks revealed that the collected parts of the micrographs used in the dataset had a significant impact on the

improvement in accuracy and reduction in loss. In particular, the performance of the neural networks tended to improve slightly during training with the cross-sectional micrographs of latewood. This phenomenon could be attributed to the anatomical characteristics of the latewood cross-section in the dataset. Clear differences in cell structure were observed between the latewood samples from the 10 species in the dataset. The latewood cross-section exhibited more advantageous characteristics for feature selection and extraction from a feature engineering perspective [32], such as cell wall thickness, appearance of the growth ring boundary, transition from earlywood to latewood, and epithelial cells. In contrast, the diameter and lumen area of the tracheids, presence of axial parenchyma, and width and frequency of the ray tissue in the earlywood cross-section were less helpful anatomical features for wood classification. Therefore, the latewood part enabled a relatively clear classification compared to the earlywood, resulting in improved accuracy and reduced loss.

The augmented dataset in the training process showed a more stable tendency in terms of accuracy and loss fluctuation with increasing epochs compared with the non-augmented dataset. Dataset augmentation prevents overfitting and improves accuracy [33–35]. In the present study, it can be concluded that the augmented dataset, which was generated by rotating, vertically and horizontally shifting, zooming in and out, and flipping the micrographs composing the dataset vertically and horizontally, contributed to the performance improvement of the four architectures.

## 5. Conclusions

The results of this study confirmed that the four convolutional neural network architectures could classify 10 sample species with an accuracy of over 90%. Factors such as epochs, total and latewood datasets, and dataset augmentation affect the accuracy of species classification. Epochs, latewood datasets, and augmented datasets improved the classification accuracy. In comparison of the average accuracy in the final five epochs, the latewood dataset exhibited a 5.8% and a 12.1% higher accuracy than the earlywood dataset and the total part dataset, respectively. In addition, the augmented dataset showed an 8.0%, 2.7%, and 2.3% higher accuracy in the total, earlywood, and latewood datasets compared to the non-augmented dataset, respectively.

The learning process in the augmented dataset for training was faster and more stable than that in the non-augmented dataset. To achieve more stable performance for wood species classification using CNNs, it is necessary to use augmented latewood datasets. The factors for performance improvement were verified with Pearson correlation coefficients of 0.289 ** in the latewood dataset and 0.172 ** in the augmented dataset.

During dataset construction, the variability of the classification accuracy and loss decreased with an increase in the number of microscope images, leading to more stable results. It is possible to classify more wood species by expanding the size of the dataset.

In the present study, epochs, latewood datasets, and augmented datasets for improving species classification accuracy using artificial neural networks are suggested to be helpful in developing a more accessible automated species identification system in the future.

**Author Contributions:** Conceptualization, J.-H.K. and N.-H.K.; methodology, J.-H.K. and N.-H.K.; software, J.-H.K.; validation, S.Y-.H., S.-H.L. and N.-H.K.; formal analysis, J.-H.K.; investigation, J.-H.K., S.-Y.H., B.D.P., D.P. and G.-U.Y.; resources, N.-H.K.; data curation, J.-H.K. Writing—original draft preparation, J.-H.K.; writing—review and editing, J.-H.K., A.M.S., B.D.P., D.P., S.-Y.H., S.-H.L. and N.-H.K.; visualization, J.-H.K.; supervision, N.-H.K.; project administration, J.-H.K. Funding acquisition: S.-H.L. and N.-H.K. All authors have read and agreed to the published version of the manuscript.

**Funding:** This research was supported by the Science and Technology Support Program through the National Research Foundation of Korea (NRF) funded by the Ministry of Science and ICT (MSIT) (No. 2022R1A2C1006470), the Basic Science Research Program through the NRF funded by the Ministry of Education (No. 2018R1A6A1A03025582), and the R&D Program for Forest Science Technology (Project Nos. 2021350C10-2323-AC03 and 2021311A00-2122-AA03) provided by the Korea Forest Service (Korea Forestry Promotion Institute).

**Data Availability Statement:** The datasets used and/or analyzed in the current study are available from the corresponding author upon reasonable request.

**Conflicts of Interest:** The authors declare no conflict of interest.

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
