# Peer review of "Performance Influencing Factors of Convolutional Neural Network Models for Classifying Certain Softwood Species"

_forests, doi:10.3390/f14061249_

Round 1
Reviewer 1 Report
The paper presents an interesting study.
1) Architecture of CNN needs to be provided.
2) Results need to be discussed in details. Proper reasoning should be provided for all the results obtained.
3) Conclusion needs to be supported with numerical values.
4) Comparative results in a single table or graph must be provided for the 4 CNNs used. Also, their comparative performance needs to be discussed thoroughly.
N/A
Author Response
Dear Reviewer,
Thank you for your review this manuscript.
According to your suggestion, we revised the four point.
1) Insert the CNN architecture(figure 2)
2) Enhencement of discussion
3) Insert the quantitative analysis in conclusion
4) Comparison among the four architecture(Table 9 and Table 11)
Thank you.

Reviewer 2 Report
The purpose of this research is to validate the wood classification performance of convolutional neural networks (CNNs) such as VGG16, ResNet50, GoogLeNet, and basic CNN architectures, as well as to analyze the factors influencing classification performance.
For me, it is not clear which performance factors exactly, that you investigated. The statement "an augmented latewood dataset was the most important factor affecting the classification performance and should be used for training CNNs." is already known in the computer vision field. It is valid for different kinds of image datasets and has already been proven in numerous studies. The title gives the expectation to the reader, that the focus will be on the performance factors, which is not the case of the introduction. It will be good to construct the article in a manner, that is related to a list of performance factors, that you will investigate.
It will be good to place Figure 2 on one page, as well as Table 6, Figure 5, Table 9, Table 4, and Table 3.
Author Response
Dear Reviewer,
Thank you for your review this manuscript.
According to your suggestion, we revised the few point.
1) Change the ambiguity terms in title as "Performance Factor Affecting -" into "Performance Influencing Factors -" for reducing confusion.
2) We tried to find the general results of wood species classification by computer vision. However, we couldn't discover the statistical analysis regarding performance influencing factors. And, No one tried to compare the collected part such as total part, earlywood, and latewood. Thus, we found some hope that this study could be helpful to someone.
3) We revised the figures and tables into placed on one page.
Thank you.

Round 2
Reviewer 1 Report
Accept
Reviewer 2 Report
Still there are spliitted figures and tables.
"2.2.2. Dataset pretreatment" need to be 2.2.2. Dataset preprocessing